# Antibiofilm Potential of Coelomic Fluid and Paste of Earthworm *Pheretima posthuma* (Clitellata, Megascolecidae) against Pathogenic Bacteria

**DOI:** 10.3390/microorganisms11020342

**Published:** 2023-01-30

**Authors:** Mudassar Hussain, Iram Liaqat, Urooj Zafar, Sadiah Saleem, Muhammad Nauman Aftab, Awais Khalid, Yosra Modafer, Fahdah Ayed Alshammari, Abdullah Mashraqi, Ahmed A. El-Mansi

**Affiliations:** 1Microbiology Lab, Department of Zoology, Government College University, Lahore 54000, Pakistan; 2Department of Microbiology, University of Karachi, Karachi 75270, Pakistan; 3Institute of Industrial Biotechnology, Government College University, Lahore 54000, Pakistan; 4Department of Physics, Hazara University Mansehra, Mansehra 21300, Pakistan; 5Department of Biology, College of Science, Jazan University, Jazan 45142, Saudi Arabia; 6Department of Biology, Faculty of Science and Arts—RAFHA, Northern Border University, Rafha 73213, Saudi Arabia; 7Department of Biology, College of Science, Jazan University, Jazan 82817, Saudi Arabia; 8Department of Biology, Faculty of Science, King Khalid University, Abha 62529, Saudi Arabia; 9Department of Zoology, Faculty of Science, Mansoura University, Mansoura 35516, Egypt

**Keywords:** *Pheretima posthuma*, coelomic fluid, body paste, time kinetics, antibiofilm

## Abstract

Antibiotic drug resistance is a global public health issue that demands new and novel therapeutic molecules. To develop new agents, animal secretions or products are used as an alternative agent to overcome this problem. In this study, earthworm (*Pheretima posthuma*) coelomic fluid (PCF), and body paste (PBP) were used to analyze their effects as antibiofilm agents against four bacterial isolates MH1 (*Pseudomonas aeruginosa* MT448672), MH2 (*Escherichia coli* MT448673), MH3 (*Staphylococcus aureus* MT448675), and MH4 (*Klebsiella pneumoniae* MT448676). Coelomic fluid extraction and body paste formation were followed by minimum inhibitory concentrations (MICs), biofilm formation time kinetics, and an antibiofilm assay, using heat and cold shock, sunlight exposure auto-digestion, and test tube methods. The results showed that the MIC values of PCF and PBP against *S. aureus*, *P. aeruginosa*, *K. pneumoniae*, and *E. coli* bacterial isolates ranged from 50 to 100 μg/mL, while, the results related to biofilm formation for *P. aeruginosa*, *S. aureus*, and *K. pneumoniae* strains were observed to be highly significantly increased (*p* < 0.005) after 72 h. *E. coli* produced a significant (*p* < 0.004) amount of biofilm after 48 h. Following time kinetics, the antibiofilm activity of PCF and PBP was tested at different concentrations (i.e., 25–200 μg/mL) against the aforementioned four strains (MH1–MH4). The findings of this study revealed that both PBP (5.61 ± 1.0%) and PCF (5.23 ± 1.5%) at the lowest concentration (25 μg/mL) showed non-significant (*p* > 0.05) antibiofilm activity against all the selected strains (MH1-MH4). At 50 μg/mL concentration, both PCF and PBP showed significant (*p* < 0.05) biofilm inhibition (<40%) for all isolates. Further, the biofilm inhibitory potential was also found to be more significant (*p* < 0.01) at 100 μg/mL of PCF and PBP, while it showed highly significant (*p* < 0.001) biofilm inhibition at 150 and 200 μg/mL concentrations. Moreover, more than 90% biofilm inhibition was observed at 200 μg/mL of PCF, while in the case of the PBP, <96% biofilm reduction (i.e., 100%) was also observed by all selected strains at 200 μg/mL. In conclusion, earthworm body fluid and paste have biologically active components that inhibit biofilm formation by various pathogenic bacterial strains. For future investigations, there is a need for further study to explore the potential bioactive components and investigate in depth their molecular mechanisms from a pharmaceutical perspective for effective clinical utilization.

## 1. Introduction

Clinically significant antibacterial agents have encountered an exponential rise in multidrug resistance over the last two decades due to the pragmatic approach taken by physicians to overcome antibacterial resistance and the accompanying patient preference for over-the-counter antibacterial drugs [1]. Antibacterial resistance has been described by a diverse range of mechanisms; for example, (a) by modifying the antibacterial drug target through post-translational alterations or genetic mutations, (b) by deactivating the antibacterial agents through modification or hydrolysis, for example, phosphorylation by the kinase enzyme, (c) by increasing the efflux of the antibiotic out of the bacterial cell through efflux pumps and porins, and (d) by decreasing the influx/penetration of the antibacterial agents into the bacterial cell through changes in the cell wall structure [2]. However, the development of biofilms is one of the favored and frequently used approaches to counteract the effects of antibacterial agents by bacteria [3].

Over 80 to 90% of pathogenic bacterial strains, including *Pseudomonas aeruginosa*, *Staphylococcus aureus*, *Escherichia coli*, and *Klebsiella pneumoniae*, possess an innate ability to form biofilms, which make biofilms the main contributor in multidrug resistance among bacteria [4]. Bacteria residing in biofilms exhibit a significantly higher pattern of adaptive resistance against antibacterial drugs and other disinfectants compared to their planktonic components. The worldwide increase of adaptive antibacterial resistance acts as a barrier when treating biofilm-associated infections, for example nosocomial and ventilator-associated pneumoniae, catheter-associated infections, surgical wound infections, burn wound infections, etc., [5]. In order to combat the bacterial and biofilm developments, new approaches need to be devised in addition to antibiotics and antibacterial drugs. In the last two decades, natural animal and plant compounds have been extracted and reported as novel solutions to avoid biofilm development [6]. 

Earthworms have also been utilized in medicines for the treatment of various diseases since 1340 AD. We recently published that the defense system of earthworms is highly complicated because they lack antibodies in their blood circulatory system. However, their body extracts and coelomic fluid comprise of different bioactive agents (i.e., peptides and proteins) that defend these worms. Among the bioactive agents, proteases, metabolites, metal-binding proteins, active proteins (include lysozyme, lysenin, and eiseniapore), (lumbricin I, lumbricin PG, and lumbricusin), organic acids, and organic compounds, confer therapeutic potential [7], nodule encapsulation and production, phagocytosis [8] and wound healing [9], while humoral responses are triggered by proteases, lysozymes, opsonins, cytolytic proteins, haemolysins, lectin-like, agglutinins, motif recognition agents, and antibacterial bioactive proteins [10]. Similarly, the invading microbes can be removed in different ways. For example, (a) they can be expelled through nephridia or captured by nephrostome cells, (b), the microbes can be engulfed by some phagocytic cells and coelomocytes and excreted out (when they exhausted) by dorsal pores, and (c) big size foreign objects (i.e., agglutinated parasites or bacteria) can be released by encapsulation [11].

The use of earthworms as medicine for the treatment of various diseases is a very ancient practice [12]. For example, in Burma, China, and India, earthworms are widely used for curing certain illnesses such as pyorrhea, postpartum weaknesses, arthritis, inflammation, wound healing, gastric ulcer, hepatoprotective, carcinoma, antipyretic, and nervous system disorders [12]. Earthworms’ tissue extracts, coelomic fluid, and body paste possess various agents (i.e., proteins) that have been well-documented as antiulcer [13], antiviral [14], anti-coagulant [15], antifungal [16], antibacterial [13], anti-inflammatory [17], antitumor [18], antipyretic [19], analgesic agents [20], and cytotoxic [21]. 

Despite the large collections of bioactive agents, there is no evidence of earthworm utilization as antibiofilm- and antiquorum-sensing agents to date. Additionally, among the Megascolecidae family members, *P. posthuma* was observed as the most abundant earthworm species in Pakistan. Owing to the relative abundance of *P. posthuma* in Pakistan, the current study utilized it as the experimental organism and assessed the biofilm formation time kinetics and antibiofilm effects of *P. posthuma* coelomic fluid (PCF) as well as body paste (PBP) against pathogenic bacteria.

## 2. Material and Methods

### 2.1. Earthworm Collection

Earthworms were collected by digging up soil at 35 cm depth with scraper from bank sides of Marala Ravi Link (MRL) Canal near village, Kanwanlit (32°20′54.5″ N 74°25′34.1″ E), in the Sialkot region, from July to September 2019. Moist, shady, grassy, and pesticide-free land sites were selected for sampling, which were confirmed by earthworms’ casting [22]. Mature earthworms were hand sorted on the basis of their presence and the absence of their clitellum. For identification, these worms were transported in soil to microbiology laboratory in the Department of Zoology, GC University Lahore. Earthworms were morphologically identified on the basis of their skin color, clitellum shape, number of segments, and prostomium under binocular microscope, using the figures and keys designed by Blakemore [23].

### 2.2. Coelomic Fluid Extraction

After depuration, *P. posthuma* earthworms were rinsed and cleaned with ddH_2_O (double-distilled water) to remove debris from the worms’ body surface. Further, these earthworms were transferred onto dry filter papers to absorb water from the skin surface. Following cleaning, heat and cold shock methods were performed to collect *P. posthuma* coelomic fluid (PCF). In brief, 30 g of earthworms were weighed and transferred into a conical funnel that was fixed on iron stand. A clean, dry nylon mesh with small pores was employed and fixed in this conical funnel to avoid the escape of earthworms.

During heat shock method, 25 mL hot water (45 ± 1 °C temperature) in a heat resistant plastic bag was used to give heat shocks for 4 min [24]. The hot water stimulated the dorsal pores in the epidermis of the earthworms to release PCF [25]. The heat shocks technique yielded 0.6 ± 0.2 mL of PCF. After 10 min gap, which overcomes the heat shock effect, earthworms were treated with cold shock. In this treatment, a bag of ice pieces was placed over the earthworms to give ice cold shock. The ice pieces decreased the temperature of earthworms’ epidermis to 15 ± 1 °C and stimulated the earthworms to secrete PCF, which was collected using 15 mL sterilized beaker. The PCF was then shifted to sterilized polypropylene tubes using disposable dropper. After harvesting, PCF was immediately centrifuged at 14,000 rpm for 14 min to get rid of impurities and contaminants. Further, supernatant was then percolated through 0.2 µm (pore size) membrane filter syringe to obtain cell-free PCF. The cell-free filtrate of PCF was stocked in sterile Eppendorf at −20 °C until further use [26].

### 2.3. Body Paste Formation

Following PCF formation, 30 g of fully mature clitellated *P. posthuma* (earthworms) were unsoiled with hands and were washed under running tap water and transferred onto wet blotting paper for 48 h to evacuate undigested fecal matter from alimentary canal. To avoid any contamination, the earthworms were again, cleaned under running tap water. The earthworms were shifted to plastic container and tightly covered with porous polythene sheet to reduce the chances of escape. After the covering, containers were kept in sunlight for 3 days to kill the worms. During this period, the secreted mucous and coelomic fluid digested the tissues of dead earthworms and formed a brown-colored body paste called *P. posthuma* body paste (PBP) [19]. The PBP was then filtered, and the obtained filtrates were condensed at 36 °C in water bath. The crude PBP acquired was diluted by adding 10% DMSO (Sigma-Aldrich, Germany) for analyses of antibiofilm potential [27].

### 2.4. Pathogenic Bacteria Collection

For antibiofilm assay, four bacterial isolates MH1 (*Pseudomonas aeruginosa* MT448672), MH2 (*Escherichia coli* MT448673), MH3 (*Staphylococcus aureus* MT448675), and MH4 (*Klebsiella pneumoniae* MT448676) were collected from microbiology lab, Department of Zoology, Government College University Lahore, Pakistan. 

### 2.5. Minimum Inhibitory Concentrations (MICs)

Micro-dilution method was utilized to examine MICs using fresh bacterial broth culture following method designed by Augustine et al. [21]. Two-fold serial dilutions of PCF were produced to obtain final dilution concentrations range between 0.781 to 400 µg/mL. A total of 200 µL of fresh bacterial Mueller–Hinton broth was poured into three columns of 96-well micro-test microplates for each pathogenic bacterial strain. So that, each well consisted of 50 µL of bacterial inoculum and 150 µL of fresh nutrient broth. Each well was then supplemented with different PCF concentrations, except for control. After covering with aluminum foil, the 96-well microplates were placed in an incubator (Vevor, JTD-8000) at 37 °C for 24 h. The PCF concentrations that lacked visible bacterial growth (i.e., turbidity) were measured as MICs. All treatments were performed in triplicates, and the mean values of MICs were calculated. Similarly, PBP concentrations were examined to find MICs against same bacterial strains.

### 2.6. Biofilm Formation Time Kinetics

The time kinetics of biofilm formation were assessed by employing the procedure as developed by Liaqat et al. [28] with some modifications. Briefly, the selected pathogenic bacterial strains were inoculating in 5 mL of sterile fresh LB broth and placed in an incubator at 37 °C for various time periods (i.e., 0, 24, 48, 72, 96, and 120 h). The biofilm formation was monitored using 0.1% of the crystal violent (CV) staining method. Meanwhile, the overnight fresh cultures of the pathogenic bacterial strains were developed and diluted, and the optical density (OD_594_) was adjusted for each strain to attain 1 × 10^8^ CFU (colony-forming unit)/mL using spectrophotometer (UV-1700, Shimadzu, Kyoto, Japan). Next, 50 μL of diluted fresh cultures was added into sterile test tubes possessing 5 mL of fresh LB media and was placed in shaking incubator at 37 °C for various time intervals, i.e., 0, 24, 48, 72, 96, and 120 h. Afterwards, the medium was discarded and washed 3× using ddH_2_O. Later, these test tubes were air-dried in inverted form under laminar flow (BBS-V1300, Biobase, Jinan, China) for 15 min at room temperature. Five mL of 0.1% CV stain was mixed in each test tube for 20 min and tubes were rinsed with 0.85% saline solution. Later, 5 mL of 33% anhydrous acetic acid (also called glacial acetic acid) was poured into tubes to dissolve the adherent-stained bacterial cells, and OD_594_ was monitored using spectrophotometer. The experiment was run in triplicates.

### 2.7. Antibiofilm Assay of PCF and PBP

Antibiofilm activity of PCF and PBP was assessed using test tube method with some modifications [29]. To examine this activity, 150 µL of pre-inocula of selected pathogenic strains was added in the separate disposable glass test tubes containing 5 mL of sterilized fresh nutrient broth. Following that, different concentrations of PCF (i.e., 25, 50, 100, 150, and 200 µg/mL) were added in these test tubes and placed in an incubator for 24 h at 37 °C, except for the control. After incubation, all the test tubes were unloaded and washed 3× with saline solution to eliminate unadhered planktonic bacteria. Meanwhile, the adherent bacterial strains that form biofilms on the walls of test tubes were stained using 0.1% CV, and test tubes were allowed to air dry by placing them in laminar flow. The tubes were then cleaned 3× with 33% anhydrous acetic acid to remove any planktonic cells. At the end, the absorbance was noted at OD_594_ nm, which revealed the biofilm formation ability of the selected isolates. The experiments were run 3× for all the selected bacterial strains. Similar procedure was adapted for PBP to assess its antibiofilm potential. The biofilm inhibition percentages were calculated and recorded using formula as given below:



$$\%\;\text{Biofilm inhibition}\;=1-\frac{\mathrm{OD}\;\left(\mathrm{nm}\right)\;\mathrm{of}\;\mathrm{bacteria}\;\mathrm{treated}\;\mathrm{with}\;\mathrm{PCF}\;\mathrm{or}\;\mathrm{PBP}\;\mathrm{Concenrations}}{\mathrm{OD}\;\left(\mathrm{nm}\right)\;\mathrm{of}\;\mathrm{control}\;\left(\mathrm{untreated}\right)}\times 100$$



### 2.8. Statistical Analysis

The values of all the time kinetic analyses and antibiofilm assays for both PCF and PBP were analyzed with control using IBM SPSS Statistics (version 20.0). The whole data were taken in triplicates and expressed as mean ± SEM (standard error of the mean). Statistical comparison was made for all PCF and PBP assays with negative control utilizing ANOVA (analysis of variance), followed by post hoc Tukey’s test. The results were considered statistically significant (*p* < 0.05), highly significant (*p* < 0.01), and most significant (*p* < 0.001).

## 3. Results

Both the heat shock and cold shock methods were found to have a remarkable influence on the *P. posthuma* earthworm body to release PCF. In the case of the heat shock method, 3 g of earthworms died, while 30 g of the *P. posthuma* worms remained active and alive. These *P. posthuma* worms secreted 1.1 ± 0.5 mL of PCF during 25 min of heat shock (45 ± 1 °C). The collected PCF appeared to be apparently brown in color, with little debris contamination, while in the case of the cold shock method, the *P. posthuma* worms were exposed to 15 ± 1 °C for 25 min, resulting in 1.8 ± 0.4 mL of PCF collection without any death of the worms. After that, all the earthworms were observed active and alive throughout the shock period. So, it was considered a relatively more safe and effective method due to the larger volume of PCF collection without any mortality and debris contamination. Both the experiments were repeated three times within a time period of 72 h. There was no significant reduction in the PCF volume during the collection methods (Table 1).

PBP was formed by the auto-digestion of the earthworms’ body tissue through mucous and coelomic fluid under sunlight. The color of PBP was dark brown and had debris contaminants. A total of 10% DMSO (dimethyl sulfoxide) was used to dilute the crude PBP and filter it. The filtrate was clear and slightly brown in color. The PBP and PCF were studied further to analyze them as antibiofilm agents.

The results regarding the MICs of PBP and PCF against four bacterial isolates were shown in Table 2. The MICs of PBP and PCF against *S. aureus*, *P. aeruginosa*, *K. pneumoniae*, and *E. coli* bacterial isolates ranged from 50 to 100 µg/mL. The minimum concentration of PCF and PBP against *E. coli* bacterial strains was 50 µg/mL, while it was 100 µg/mL against *P. aeruginosa*. Further, 100 µg/mL of concentrations for both PBP and PCF were observed as sufficient to inhibit the growth of four selected bacterial isolates. 

The results related to biofilm formation time kinetics by the MH1 strain were assessed and observed as highly significant (*p* < 0.005; 0.397 ± 0.02) at a 72-h incubation period. Similarly, biofilm was also detected as high (0.352 ± 0.03) at a 96-h incubation period, but it was significantly lower than the optimal value at a 72-h incubation period. Similarly, MH2 developed significant (*p* < 0.004; 0.378 ± 0.02) levels of biofilm at 48 h of incubation. Moreover, the results of both MH3 and MH4 showed significantly increased biofilms (392 ± 0.02 and 0.348 ± 0.03) after a 72-h incubation at 37 °C. From the results, it was concluded that biofilm formation was highest at 72 h for MH1, MH3, and MH4. In contrast to these, an optimal level of biofilm in MH2 was detected at 48 h of incubation. Afterwards, a decline phase was observed (Figure 1 and Table 3).

The antibiofilm potentials of both PBP and PCF at various concentrations, 25, 50, 100, 150, and 200 µg/mL, were also checked against MH1, MH2, MH3, and MH4. All selected strains cultures (MH1, MH2, MH3, and MH4) showed a non-significant (*p* > 0.05) inhibition of the biofilm at a 25 µg/mL concentration of both the PBP (0.3 ± 0.1 to 5.61 ± 1.0%) and PCF (0.8 ± 0.33 to 5.23 ± 1.5%). As these inhibitory values (<10%) were found to be very low, no biofilm inhibition was found compared to the untreated control. Moreover, less significant (*p* < 0.05) results were observed at a 50 µg/mL concentration of PBP (except for MH1 and MH2, which showed >50%) and PCF against all selected mono-strain cultures since <50% biofilm inhibition was found (Figure 2, Table 3).

Further, a 100 µg/mL concentration of PBP and PCF exhibited a highly significant (*p* < 0.01) reduction (except for PCF for MH3, *p* < 0.05) of the biofilm by all mono-cultures. Interestingly, the results also revealed that a 63 to 90% inhibition of biofilm formation was observed at 150 µg/mL concentrations for both PBP and PCF against all selected cultures, and that the values were found more significant (*p* < 0.01) compared to the control, while the maximum inhibition (i.e., 90 to 99.8%) was found at the highest concentration (200 µg/mL) for both PBP and PCF (*p* < 0.001). Hence, it was noted that PBP exhibited more significant inhibitory potential than PCF. Moreover, the maximum biofilm inhibition of PBP and PCF was measured as 99.8 ± 1.0% against MH4 strains and 95.91 ± 1.33% against MH2 strains, respectively. Overall, the results established a directly proportional relationship between the PBP and PCF doses and the biofilm inhibition percentage (Figure 2 and Table 4). 

## 4. Discussion

Biofilm-forming bacteria are innately resistance to antibacterial agents (i.e., antibiotics), which constitutes the leading cause of numerous infectious diseases in animals and humans [30]. Nowadays, the discovery of harmless antibacterial and anti-biofilm compounds has received a lot of attention because such molecules would not eventually result in drug resistance [31]. For example, plant and animal extracts with a variety of bioactive agents are strong contenders for the treatment of different infections caused by the bacterial biofilms [32]. In this study, the antibiofilm activities of *P. posthuma* coelomic fluid (PCF) and paste (PBP) were assessed against human pathogenic bacteria.

The results indicated that PCF and PBP of *P. posthuma* inhibited bacterial growth with the highest MIC values against *P. aeruginosa*, while the lowest MIC values were observed against *E. coli*. However, generally the MICs of PCF and PBP ranged between 50 to 100 µg/mL against both the *S. aureus* and *K. pneumoniae* strains. While comparing the data, it was observed that the MIC values recorded in the current study were lower than the values obtained by other researchers who reported the highest MICs (200 µg/mL) of earthworm extracts against *E. coli* [31], *S. aureus* and *K. pneumoniae* [33]. The difference might be due to the variations in the media used and the selected strains, as previously noted [34]. 

The most widely employed method for analyzing biofilm formation time kinetics and antibiofilm potential used a 96-well microtiter plates assay [35]. Results regarding the biofilm formation time kinetics indicated that all four bacterial isolates MH1 (*P. aeruginosa*), MH2 (*E. coli*), MH3 (*S. aureus*), and MH4 (*K. pneumoniae*) formed the optimal level of biofilm at 48 to 72 h of incubation. Previously, various researchers also used the crystal violet method as the standard to measure biofilms developed by different bacteria [36]. The results of this study corroborate the findings of Liaqat et al. [37], who reported that *P. aeruginosa* started to synthesize biofilm after 24 h of incubation but produced a strong biofilm at 72 h of incubation. Our results also sustained that *E. coli* formed a strong biofilm at 48 h of incubation [38], while the findings by Yulinery et al. [39] showed biofilm development after 3 days (72 h) of incubation, with an increased number of *E. coli* cells. Previously, researchers found the 24–72 h of the incubation period necessary for achieving the optimal levels of biofilm formation, depending upon the type of bacterial strains used, culture conditions, media types, and adhesion patterns of bacterial species on a glass or plastic surface, etc. [40]. Additionally, researchers have showed the importance of pili for facilitating bacterial adhesion [37]. 

For *S. aureus* biofilm time kinetics, our finding co-related with the results of Oliveira et al. [40], who found a liner relationship between biofilm development and time period. The authors demonstrated that an increased time period caused high *S. aureus* biofilm formation up to 34.6, 69.2, and 80.8%, after 1-, 2- and 3-days incubation period, respectively. In the case of *K. pneumoniae*, our results showed a strong biofilm after 72 h of incubation period. This is against the findings by Singla et al. [41], who reported that *K. pneumoniae* formed a biofilm after a 5-day incubation period, after which the biofilm formation declined. Similar to our findings, Liaqat et al. [42] also observed that *K. pneumoniae* developed the highest level of biofilm after a 48-h incubation period. However, the biofilm development decreased after a 72-h incubation. Likewise, Toyofuku et al. [43] also found that, depending upon the type of bacterial strains, certain environmental factors, such as pH, temperature, nutrients concentration, and adherent surface, also contribute to biofilm time kinetics.

The antibiofilm activity of PBP and PCF (25 to 200 µg/mL) was analyzed against MH1 (*P. aeruginosa*), MH2 (*E. coli*), MH3 (*S. aureus*), and MH4 (*K. pneumoniae*), using LB broth culture and the crystal violet method [40]. The results of the antibiofilm assay were non-significant at a low concentration (25 µg/mL) for both PBP and PCF. Moreover, the findings of the current study revealed a 60–90% inhibition at 150 µg/mL, and a maximum inhibition (i.e., 80–100%) at 200 µg/mL. This showed that PBP exhibited more significant antibiofilm activity compared to PCF. At the end, it was observed that the results established a directly proportional relationship between PBP and PCF concentrations and the biofilm inhibition percentage. Similarly, Mustafa et al. [44] used the earthworms’ extract to check its effect on biofilm inhibition, cell proliferation, and the synergistic effect. They found that the earthworm extract inhibited the biofilm synthesis compared to the control. In another study, an earthworm extract (*A. cortices*) significantly decreased the biofilm production for *K. pneumoniae* and *S. pyogenes*. Moreover, the extract of *A. gracilis* also reduced the biofilm synthesis for *S. aureus* and *S. pyogenes*, while the extract of *P. posthuma* inhibited the biofilm formation for *E. coli* and *K. pneumoniae.* Likewise, body extracts of *E. fetida* reduced the biofilm development for *P. aeruginosa* [44]. Biofilm formation is based on various factors and their correlations, such as adhesions, growth, extracellular-binding proteins in matrix, biofilm architecture, and cell to cell communication, which play a vital role in formation as well as maturation [33]. 

Likewise, Bais et al. [45] extracted coelomic fluid from *Holothuria scabra* (a non-Annelida invertebrate) and used it as an antibacterial and antibiofilm agent. Their findings showed that the coelomic fluid inhibited the biofilm at a 25% concentration, which may be due to low-molecular weight antibacterial peptides, such as holothuroidin 2 [46]. In another finding, Shajani et al. [47] evaluated the antibacterial and antibiofilm potential of coelomic fluid extracted from *Echinometra mathaei* (a non-Annelida invertebrate) and found that the coelomic fluid completely inhibited bacterial biofilm at doses higher than 12.5 mg/mL, while it eradicated the biofilm at a 100 mg/mL concentration. 

Our results are in agreement with the findings of Roubalová et al. [11], who showed that earthworm’s PCF and PBP consist of many bioactive agents that control numerous biological activities, thus they might act as antibacterial and antibiofilm agents. Similarly, Hussain et al. [17] described that earthworm body paste, extract, and coelomic fluid all possess various kinds of bioactive agents, including proteases (proteins or peptides), metabolites, metal-binding proteins, active proteins (such as lysenin, lysozyme, and eiseniapore, etc.), antimicrobial peptides, coelomic cytolytic factor (CCF and CCF-I), lysenin, fetidin, lumbricin complex, organic acids (i.e., fatty acid), and some organic compounds (e.g., vitamin D and purine and vitamin D) [7,44]. These bioactive compounds might have played a significant role as antibiofilm agents in the current study.

## 5. Conclusions

The current study investigated earthworm (*Pheretima posthuma*) coelomic fluid and body paste as antibiofilm agents against four bacterial isolates: MH1 (*P. aeruginosa* MT448672), MH2 (*E. coli* MT448673), MH3 (*S. aureus* MT448675), and MH4 (*K. pneumoniae* MT448676). Among the three strains, *P. aeruginosa*, *S. aureus*, and *K. pneumoniae* showed increased biofilm formation after 72 h, compared to *E. coli*, which showed biofilm formation after 48 h. Antibiofilm activity of PCF and PBP at a 50 µg/mL concentration showed a significant (*p* < 0.05) biofilm inhibition (<40%). Further, biofilm inhibitory potential was also found to be more significant (*p* < 0.01) at 100 µg/mL of PCF and PBP against all strains, while it showed highly significant (*p* < 0.001) biofilm inhibition at 150 and 200 µg/mL concentrations. Moreover, more than 90% of biofilm inhibition was found in all isolates at 200 µg/mL of PCF. In brief, earthworm body fluid and paste have biologically active components, as we reported recently [7], which might be responsible for biofilm inhibition in a concentration-dependent manner. For the future, there is a need for further studies to extract the potential bioactive components and investigate their molecular mechanisms in depth from a pharmaceutical perspective for effective clinical utilization.

## Figures and Tables

**Figure 1 microorganisms-11-00342-f001:**
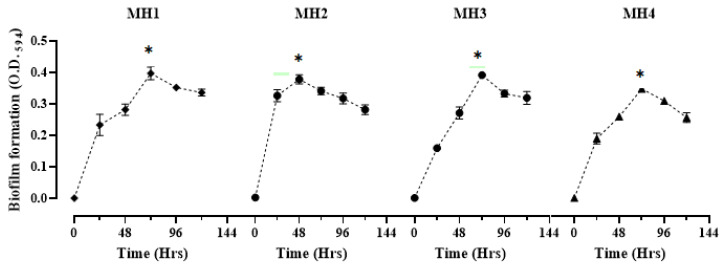
Biofilm time kinetics of MH1, MH2, MH3, and MH4 (Table 2). Bacterial strains were incubated for 0, 24, 48, 72, 96, and 120 h. Biofilm formation was examined using crystal violet (CV) staining method. Values represent mean + SD (*n* = 3) at significance level (*p* < 0.05). * and green line indicate the significantly highest biofilm formation with different time kinetics for various bacterial isolates.

**Figure 2 microorganisms-11-00342-f002:**
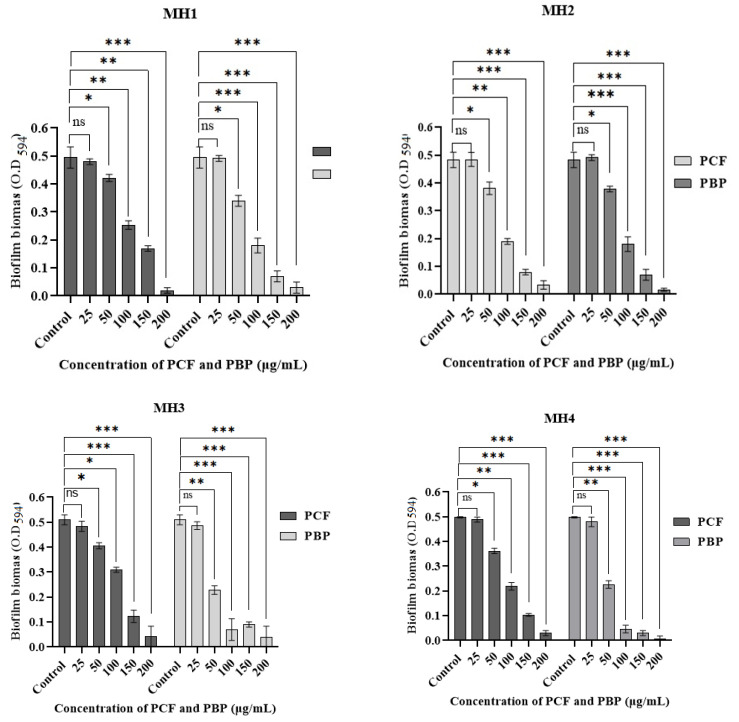
The antibiofilm potential of PBP and PCF at 25, 50, 100, 150, and 200 µg/mL concentrations against MH1 (*P. aeruginosa* MT448672), MH2 (*E. coli* MT448673), MH3 (*S. aureus* MT448675), and MH4 (*K. pneumoniae* MT448676) bacterial strain cultures. Graphs were prepared on Prism GraphPad 9.0 software. Statistical analysis was conducted by using student’s *t* test on SPSS software. All experiments were run in triplicates. All values represent mean + SD (*n* = 3). Where ns used for non-significant biofilm formation, * for significant (*p* < 0.05), ** for highly significant (*p* < 0.01), and *** for very highly significant (*p* < 0.001) values of each experiment.

**Table 1 microorganisms-11-00342-t001:** Comparison of PCF collection methods based on volume, temperature, and remarks.

Methods	Weight of *P. posthuma* Earthworms (gram)	Volume of PCF (mL)	T (°C)	Time (minutes)	Remarks
Heat shock method	30	1.1 ± 0.5	45 ± 1 °C	25	3 g of earthworms died the while others remained alive and active; produced concentrated fluid.
Cold shock method	30	1.8 ± 0.4	15 ± 1 °C	25	All earthworms were active and alive throughout the shock; excreted concentrated fluid.

PCF (*P. posthuma* coelomic fluid), T (Temperature).

**Table 2 microorganisms-11-00342-t002:** Minimum inhibitory concentration of PBP and PCF against four pathogenic bacteria.

Concentrations (µg/mL)	Pathogenic Bacterial Strains							
	*S. aureus*		*P. aeruginosa*		*K. pneumoniae*		*E. coli*	
	PBP	PCF	PBP	PCF	PBP	PCF	PBP	PCF
400	−	−	−	−	−	−	−	−
200	−	−	−	−	−	−	−	−
100	−	−	−	−	−	−	−	−
50	−	+	+	+	−	+	−	−
25	+	+	+	+	+	+	+	+
12.5	+	+	+	+	+	+	+	+
6.25	+	+	+	+	+	+	+	+
3.13	+	+	+	+	+	+	+	+
1.56	+	+	+	+	+	+	+	+
0.781	+	+	+	+	+	+	+	+
MICs (µg/mL)	50	100	100	100	50	100	50	50

Note: + sign (growth of bacteria), − sign (no growth of bacteria), PCF (*P. posthuma* coelomic fluid), and PBP (*P. posthuma* body paste).

**Table 3 microorganisms-11-00342-t003:** Biofilm formation time kinetics by different bacterial isolates and combinations at various time intervals.

Bacterial Isolates	Biofilm Formation with Time Kinetics						
	0 (h)	24 (h)	48 (h)	72 (h)	96 (h)	120 (h)	*p* Value
MH1	0.001 ± 0.0 ^e^	0.233 ± 0.01 ^d^	0.282 ± 0.0 ^cd^	0.397 ± 0.02 ^a^	0.352 ± 0.03 ^ab^	0.336 ± 0.01 ^bc^	<0.005
MH2	0.003 ± 0.0 ^d^	0.327 ± 0.01 ^b^	0.378 ± 0.02 ^a^	0.342 ± 0.01 ^ab^	0.318 ± 0.02 ^bc^	0.282 ± 0.01 ^c^	<0.004
MH3	0.005 ± 0.0 ^e^	0.160 ± 0.02 ^d^	0.272 ± 0.05 ^c^	0.392 ± 0.02 ^a^	0.333 ± 0.05 ^b^	0.320 ± 0.02 ^b^	<0.008
MH4	0.002 ± 0.0 ^e^	0.190 ± 0.01 ^d^	0.259 ± 0.02 ^c^	0.348 ± 0.03 ^a^	0.310 ± 0.04 ^b^	0.257 ± 0.01 ^c^	<0.006

Data was analyzed using one way ANOVA followed by post hoc Turkey test. No common superscripts show significant difference at *p* < 0.05.

**Table 4 microorganisms-11-00342-t004:** % of the biofilm inhibition by different bacterial isolates and their combinations at various concentrations of PCF and PBP.

Bacterial Strains	% of Biofilm Inhibition									
	Concentrations of PCF (µg/mL)					Concentrations of PBP (µg/mL)				
	25	50	100	150	200	25	50	100	150	200
MH1	0.8 ± 0.3	22.08 ± 2.6	61.15 ± 1.0	83.64 ± 2.6	93.25 ± 2.5	0.4 ± 0.3	22.69 ± 1.3	63.19 ± 3.3	85.68 ± 2.5	93.86 ± 2
MH2	1.8 ± 0.6	23.9 ± 3.5	50.87 ± 3.5	65.63 ± 2.5	95.91 ± 1.3	0.3 ± 0.1	30.47 ± 2.6	63.19 ± 0.5	85.5 ± 3.0	93.86 ± 2
MH3	5.23 ± 1	20.1 ± 0.5	54.68 ± 2.5	75.88 ± 1.5	92.74 ± 3.5	4.7 ± 0.3	55.29 ± 1.6	72.94 ± 3.3	82.35 ± 1.0	93.34 ± 2
MH4	1.6 ± 1.0	27.71 ± 2.5	56.22 ± 1.5	79.71 ± 0.3	94.97 ± 2.5	5.61 ± 1	54.81 ± 1.6	90.96 ± 1.6	94.98 ± 2.0	99.8 ± 1.0

Where MH1 (*P. aeruginosa* MT448672), MH2 (*E. coli* MT448673), MH3 (*S. aureus* MT448675), and MH4 (*K. pneumoniae* MT448676) showed cultures of bacterial strains.

## Data Availability

All the data are available in the manuscript.

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
