# Peer review of "Antibiofilm Potential of Coelomic Fluid and Paste of Earthworm Pheretima posthuma (Clitellata, Megascolecidae) against Pathogenic Bacteria"

_microorganisms, 2023, doi:10.3390/microorganisms11020342_

Round 1

Reviewer 1 Report

Although the manuscript intends to evaluate the Coelomic Fluid and Paste of Earthworm as Antibiofilm, the manuscript presented one method only to achieve this purpose without identifying the major components (in PCF and PBP) which are responsible for this matter. So, it could be appreciated as a lack of originality in the work. Also, some modifications should be addressed before it could be published.

Line 22-23; 51-58; 137-139; 153; 162; 168; 282-285; 296; 315-320; 334-335: check the font type and size.

Line 29: PCF and PBP, abbreviations should be first mention in the text and then provide the abbreviation.

Line 87 to 89: Scientific name should be Italic.

Line 104: What does ddH2O mean?

Line 111: hot water (40-50°C), line 112: 3-4 minutes, line 117: 11-15°C, and line 120: at 10000-14000 rpm. What is precisely °C, min, rpm that authors used?

Line 134-135: The crude PBP acquired was diluted by adding 10% DMSO for analyses of antibacterial potential. However, Authors did not examine the antibacterial activity in this manuscript.

Please, explain why authors used OD594 in Biofilm formation time kinetics, then used OD530 in antibiofilm assay.

In biological works, like this one, where results are influences by non-controllable effects inherent to living organisms, the use of three replicates could induce questionable results.

It would be better if authors used scientific name than Abbreviation MH... etc.

The resolution of all photos are needs to enhancement.

In table 3: 61.145±1. !!!

The discussion and conclusion parts are insufficient and presented results only. While in most cases, the result discussion is limited to compare results between them and with previous reports, but there is a lack of hypothesis of the observed behavior causes. 

Author Response

Comments and Suggestions for Authors

Although the manuscript intends to evaluate the Coelomic Fluid and Paste of Earthworm as Antibiofilm, the manuscript presented one method only to achieve this purpose without identifying the major components (in PCF and PBP) which are responsible for this matter. So, it could be appreciated as a lack of originality in the work. Also, some modifications should be addressed before it could be published.

We are thankful to Reviewers for their kind guidance. This study didn’t idnetify major ingredient found in PCF and PBP. However, we recently published a review articles with the details of all possible components conferring therapeutic potential. In addition, we also cited literature that details the medicinal importance of earthworms. All this led us to investigate the antibiofilm potential which wasn’t studied to date.

Line 22-23; 51-58; 137-139; 153; 162; 168; 282-285; 296; 315-320; 334-335:

check the font type and size.

Line 22-335, all the font type and size were checked and corrected.

Line 29: PCF and PBP, abbreviations should be first mention in the text and then provide the abbreviation.

Line 3-4; PCF and PBP, abbreviations are first mentioned in the text.

Line 87 to 89: Scientific name should be Italic.

Line 87 to 89; Earthworm P. posthuma species name is now italicized.

Line 104: What does ddH2O mean?

Line 104-105; ddH2O mean double distilled water

Line 111: hot water (40-50°C), line 112: 3-4 minutes, line 117: 11-15°C, and line 120: at 10000-14000 rpm. What is precisely °C, min, rpm that authors used?

Line 111-112: During heat shock method, 25 mL hot water (45 ± 1°C temperature) in a heat resistant plastic bag was used to give heat shocks for 4 minutes.

Line 117: The ice pieces decreased the temperature of earthworm epidermis to 15± 1°C and stimulated the earthworm to secrete PCF which was collected by using 15 ml sterilized beaker.

Line 120: After harvesting, PCF was immediately centrifuged at 14000 rpm for 14 minutes to get rid of impurities and contaminants.

Line 134-135: The crude PBP acquired was diluted by adding 10% DMSO for analyses of antibacterial potential. However, Authors did not examine the antibacterial activity in this manuscript.

Line 134-135: Yes agreed. There was no any study about antibacterial effect of PBP. Now the corrected one is “the crude PBP acquired was diluted by adding 10% DMSO (Sigma-Aldrich, Germany) for analyses of antibiofilm potential”.

Please, explain why authors used OD594 in Biofilm formation time kinetics, then used OD530 in antibiofilm assay.

It was typo mistake, only OD594 was used both in biofilm formation time kinetics and antibiofilm assay.

In biological works, like this one, where results are influences by non-controllable effects inherent to living organisms, the use of three replicates could induce questionable results.

To minimize the ambiguities, experiments were done in triplicate

It would be better if authors used scientific name than Abbreviation MH... etc.

Yes agreed. However, abbreviated names were used to minimize plagiarism. 

The resolution of all photos are needs to enhancement.

Resolution of photos was enhanced as suggested

In table 3: 61.145±1. !!!

The table 3 was corrected, and now labelled as the table 4

The discussion and conclusion parts are insufficient and presented results only. While in most cases, the result discussion is limited to compare results between them and with previous reports, but there is a lack of hypothesis of the observed behaviour causes. 

We appreciate. We have compared our results with available reported literature. However, this study is novel and first attempt as antibiofilm potential of earthworm coelomic fluid and body paste.

Reviewer 2 Report

The manuscript "Antibiofilm Potential of Coelomic Fluid and Paste of Earth worm Pheretima Posthuma (Clitellata, Megascolecidae) against Pathogenic Bacteria" has an interesting outcome. However, for publication in Microorganisms, the manuscript needs to be improved.

The discussion needs to review and compare the data with the literature. Many grammatically problematic sentences were found throughout the manuscript, which must be checked and corrected precisely by English editing services.

  1. L22-23, 137-138: Use the full form of all bacteria for the first time.
  2. L28: “P” should be capitalized and italic
  3. L87, 88, 182-183, and so on: “P. posthuma” should be italic
  4. Authors should use most updates (last 5 years) references in the introduction part.
  5. L126: delete “water” after freshwater
  6. L135, 153, and so on: Sources (manufacturer name, city, country) need to mention all the chemicals, reagents, and equipment used in this manuscript.
  7. The introduction and results sections are poor. Need to maintain a logical flow of the writing.
  8. The discussion is feeble. A sound discussion includes principal, relationships, and generalizations supported by the results.
  9. Authors should check typos throughout the manuscript.
  10. Many spacing and punctuation marks problems are found throughout the manuscript. Revision required.

Author Response

Response to Reviewer II

         English language and style

( ) English very difficult to understand/incomprehensible
(x) Extensive editing of English language and style required
( ) Moderate English changes required
( ) English language and style are fine/minor spell check required
( ) I don't feel qualified to judge about the English language and style

Yes

Can be improved

Must be improved

Not applicable

Does the introduction provide sufficient background and include all relevant references?

( )

(x)

( )

( )

Are all the cited references relevant to the research?

( )

( )

(x)

( )

Is the research design appropriate?

( )

( )

(x)

( )

Are the methods adequately described?

( )

(x)

( )

( )

Are the results clearly presented?

( )

( )

(x)

( )

Are the conclusions supported by the results?

( )

( )

(x)

( )

Comments and Suggestions for Authors

The manuscript "Antibiofilm Potential of Coelomic Fluid and Paste of Earth worm Pheretima Posthuma (Clitellata, Megascolecidae) against Pathogenic Bacteria" has an interesting outcome. However, for publication in Microorganisms, the manuscript needs to be improved.

The discussion needs to review and compare the data with the literature. Many grammatically problematic sentences were found throughout the manuscript, which must be checked and corrected precisely by English editing services.

  1. L22-23, 137-138: Use the full form of all bacteria for the first time.

L22-23, In this study, earthworm (Pheretima posthuma) coelomic fluid (PCF) and body paste (PBP) is used to analyze its effect as antibiofilm agents against four bacterial isolates MH1 (Pseudomonas aeruginosa MT448672), MH2 (Escherichia coli MT448673), MH3 (Staphylococcus aureus MT448675) and MH4 (Klebsiella pneumoniae MT448676) bacterial strains.

137-138: For antibiofilm assay, four bacterial isolates, MH1 (Pseudomonas aeruginosa MT448672), MH2 (Escherichia coli MT448673), MH3 (Staphylococcus aureus MT448675) and MH4 (Klebsiella pneumoniae MT448676) were collected from Microbiology Lab, Department of Zoology, Government College University Lahore, Pakistan.

  1. L28: “P” should be capitalized and italic

L28: All the P value are capitalized and italicized like (P < 0.001). while, results related to biofilm formation by P. aeruginosa, S. aureus and K. pneumoniae strains was observed to be highly significantly increased (P < 0.005) after 72 hours. E. coli produced significant (P < 0.004) amount of biofilm formation after 48 hours incubation.

  1. L87, 88, 182-183, and so on: “P. posthuma” should be italic

L87, 88, 182-183, and so on:  The P. posthuma is italicized  as suggested.

  1. Authors should use most updates (last 5 years) references in the introduction part.

All the old references are removed and updated literature added in place of old literature.

  1. L126: delete “water” after freshwater

L126: deleted the word water. After PCF formation, 30 g of fully mature clitellated, P. posthuma (earthworms) were unsoiled with hands and washed in running freshwater and transferred on wet blotting paper for 48 hours to evacuate undigested fecal matter from alimentary canal.

  1. L135, 153, and so on: Sources (manufacturer name, city, country) need to mention all the chemicals, reagents, and equipment used in this manuscript.

L135: The crude PBP acquired was diluted by adding 10% DMSO (Sigma-Aldrich, Germany) for analyses of antibiofilm potential.

L153: Later, these test tubes were air dried in inverted form under Laminar Flow (BBS-V1300, Biobase) for 15 minutes at room temperature.

The overnight fresh cultures of the pathogenic bacterial strains were developed, diluted and the optical density (OD) were adjusted for each strain at OD594 ± 0.1 to attain 1×108 CFU (colony forming unit)/mL using spectrophotometer (UV-1700, Shimadzu). 

  1. The introduction and results sections are poor. Need to maintain a logical flow of the writing.

We appreciated your point. We have rewritten and rephrased the introduction and results. Also tried to improve logical flow.

  1. The discussion is feeble. A sound discussion includes principal, relationships, and generalizations supported by the results.

We also improved the results by considering the critical points.

  1. Authors should check typos throughout the manuscript.

After thoroughly reading, all the typo mistakes were removed as suggested.

  1. Many spacing and punctuation marks problems are found throughout the manuscript. Revision required.

The manuscript was Carefully revised and thoroughly checked for all mistakes as suggested.

Round 2

Reviewer 2 Report

Thank you!